# Information Retention via Learning Supplemental Features

**Zhipeng Xie**[*], **Yahe Li**[*]
(* Equal Contribution)
School of Computer Science, Fudan University, Shanghai 200434, China
xiezp@fudan.edu.cn, yaheli21@m.fudan.edu.cn

## Abstract

The information bottleneck principle provides an information-theoretic method for learning a good representation as a trade-off between conciseness and predictive ability, which can reduce information redundancy, eliminate irrelevant and superfluous features, and thus enhance the in-domain generalizability. However, in low-resource or out-of-domain scenarios where the assumption of i.i.d does not necessarily hold true, superfluous (or redundant) relevant features may be supplemental to the mainline features of the model, and be beneficial in making prediction for test dataset with distribution shift. Therefore, instead of squeezing the input information by information bottleneck, we propose to keep as much relevant information as possible in use for making predictions. A three-stage supervised learning framework is designed and implemented to jointly learn the mainline and supplemental features, relieving supplemental features from the suppression of mainline features. Extensive experiments have shown that the learned representations of our method have good in-domain and out-of-domain generalization abilities, especially in low-resource cases.

## 1 Introduction

Deep neural network (DNN) has revolutionized a variety of fields (inclusive of but not limited to computer vision, natural language processing and speech recognition) in supervised learning tasks (Alam et al., 2020; Otter et al., 2020). Although usually overparameterized, DNN has shown good generalization ability and performs well on test data. Dating back to the Occam's Razor philosophical principle, it has long been believed that a good model should be as simple as possible. Some research works (Shwartz-Ziv & Tishby, 2017; Neyshabur, 2017) have been done to uncover the implicit regularization phenomenon or mechanism in the training process of DNNs.

To reveal the dynamics of DNN training process, by visualizing information plane, it was observed (Shwartz-Ziv & Tishby, 2017) that most of the DNN training efforts are spent on compressing the input to concise representation. This observation has justified the application of the information bottleneck principle (Tishby et al., 1999; Tishby & Zaslavsky, 2015) in the supervised learning settings, seeking to capture and efficiently represent the relevant information in the input variable about the label variable and building up a good presentation in terms of a fundamental trade-off between conciseness and good predictive ability. By explicitly minimizing the mutual information between the input data and its representation and simultaneously maximizing the mutual information between the representation and the label, it has been shown the information bottleneck principle has led to the robustness of fitted model (Alemi et al., 2017).

On the other hand, in the setting of unsupervised representation learning, there is no supervised label to identify relevant or irrelevant information, and thus the InfoMax principle (Linsker, 1988) is usually used to maximize the mutual information between the input and its representation (Oord et al., 2018; Hjelm et al., 2019), with the expectation that all the (potentially) predictive information is preserved for various downstream supervised tasks.

In contrast to the information bottleneck principle that ignores as many details of the input, our work goes to another end, with the so-called *Information Retention* principle as our main thrust: when making predictions (or decisions), it is preferable to keep as much relevant information as

possible in use. In other words, the principle of information retention differs from the information bottleneck: information retention explicitly preserves the relevant redundant information, but information bottleneck implicitly suppresses the redundant relevant information. Bearing the idea of information retention in mind, we design and implement a three-stage process for supervised learning. It firstly learns mainline features via vanilla DNN model, then erases salient input features (with respect to the mainline features) from the original inputs to produce modified inputs, and finally uses a conditional mutual information as regularization term forcing the supplemental features to be complementary to the mainline features and alleviating feature suppression.

In addition, our approach is also different from the InfoMax principle. It is infeasible to directly maximize the mutual information between the representation and the input in supervised learning, because it will learn irrelevant information and produce irrelevant features without any restriction.

Our contribution is three folds:

- We propose the information retention principle that favors using as much relevant information as possible in supervised learning.

- To alleviate the problem of feature suppression, we develop a three-stage process for information retention via learning supplemental features.

- Experimental results indicate that the learned representation of our method has better in-domain and out-of-domain generalization ability than several competitors.

To better motivate the idea of information retention, we also provide our thought from the perspective of causal diagram (Appendix D.1) and investigate a simple toy example (Appendix D.2).

## 2 METHOD

This section focuses on the details of the proposed method that is called "**Info**rmation **R**etention by **L**earning **S**upplemental **F**eatures" (or **InfoR-LSF** in short). The basic idea is to build up a more abundant supervised representation for classification task. To facilitate discussion, this paper assumes that a DNN-based classification model consists of two parts: the first is an encoder that maps an input vector $\mathbf{x} \in \mathcal{R}^M$ to a representation vector $\mathbf{z} \in \mathcal{R}^D$, and the second is a classification head that maps the representation vector $\mathbf{z}$ to the final prediction $\tilde{\mathbf{y}}$. Finally, the $\mathcal{L}(\tilde{\mathbf{y}}, \mathbf{y})$ denote the supervised loss used to train the model, where $\mathbf{y}$ is the true label of $\mathbf{x}$. In our method, the representation vector $\mathbf{z} = [\mathbf{z}_M; \mathbf{z}_S] \in \mathcal{R}^D$ is divided into two groups: the *mainline* representation $\mathbf{z}_M \in \mathcal{R}^{D/2}$, and the *supplemental* representation $\mathbf{z}_S \in \mathcal{R}^{D/2}$, which are learned in a three-stage process. The first stage can be thought of as a "burn-in" phase, initially building up the mainline representation from the original train data. The second stage erases the salient input features from each original input and produces a modified input, where the salient input features denote the features that are important with respect to the mainline representation $\mathbf{z}_M$. The third stage aims to jointly learn mainline representation $\mathbf{z}_M$ and supplemental representation $\mathbf{z}_S$ by forcing $\mathbf{z}_S$ to forget the salient input features that have already been captured by mainline representation and alleviate feature suppression, with the help of the modified inputs from the second stage. In the following sections, we will specifically discuss each stage below and also provide an algorithm table in Appendix C.

### 2.1 THE FIRST STAGE: INITIAL TRAINING OF MAINLINE FEATURES

At the first stage, the task is to train an initial encoder and obtain the mainline features $\mathbf{z}_M$ by maximizing the mutual information between $\mathbf{z}_M$ and the label $\mathbf{y}$ and (optionally) simultaneously minimizing the mutual information between $\mathbf{z}_M$ and input $\mathbf{x}$. Therefore, the first-stage objective is:

$$\text{maximize} \quad I(\mathbf{z}_M; \mathbf{y}) - \beta \cdot I(\mathbf{z}_M; \mathbf{x}) \tag{1}$$

where $\beta$ is the coefficient used to control information compression. Setting $\beta = 0$ will disable the information bottleneck mechanism.

Figure 1 shows the architecture of the first stage for initially training of our method. The encoding network consists of a backbone $f_\theta(\cdot)$ and a variational encoder $g_\phi(\cdot)$ with few parameters. For input $\mathbf{x}$, we use an encoding network to obtain the distribution parameters $\mu$ and $\Sigma$, and generate

$$\text{maximize} \quad I(\mathbf{z}_M; \mathbf{y}) - \beta \cdot I(\mathbf{z}_M; \mathbf{x})$$

$(\mathbf{x}, \mathbf{y}) \rightarrow$ | backbone network $f_\theta(\cdot)$ | $\xrightarrow{\mathbf{h}}$ | Variational Encoder $g_\phi(\cdot)$ | $\xrightarrow{\mu, \boldsymbol{\Sigma}}$ | RT | $\rightarrow \mathbf{z}_M \sim \mathcal{N}(\mu, \boldsymbol{\Sigma})$

Figure 1: The architecture of the first stage for initially training the mainline features.

the mainline representation $\mathbf{z}_M$ through reparameterization trick (Kingma & Welling, 2013). It should be highlighted that in the first stage, we only calculate the supervised loss with information bottleneck restriction on the mainline representation.

## 2.2 THE SECOND STAGE: SALIENCY ERASING FROM INPUTS

The objective of the second stage is to find and erase salient input features with respect to mainline features $\mathbf{z}_M$ from input $\mathbf{x}$ and produce a modified input which will be used in the third stage for auxiliary loss calculation. Formally, we denote $\mathbf{x}_{\text{sf}}$ as the most salient features that $\mathbf{z}_M$ has learned in $\mathbf{x}$, and utilize $\mathbf{x}' = \text{MASK}(\mathbf{x}) = \mathbf{x}/_{\mathbf{x}_{\text{sf}}}$ to represent the modified input after removing $\mathbf{x}_{\text{sf}}$ from $\mathbf{x}$, where $\text{MASK}(\cdot)$ refers to a feature erasure operation corresponding to the data type at hand.

**Salient Input Feature Selection.** Given an input data $\mathbf{x}$, its mainline features can be obtained as $\mathbf{z}_M = g_\phi(f_\theta(\mathbf{x}))$ via the well-trained mainline representation extractor $g_\phi(f_\theta(\cdot))$ of the first stage. The task here is to find a feature subset $\mathbf{x}_{\text{sf}} \subset \mathbf{x}$ that the current model relies on most to make prediction. To measure the importance of input features, we use the norm of the loss gradient with respect to input features.

$$\mathbf{x}_{\text{sf}} = \underset{x \in \mathbf{x}}{\text{topK}} \, ||\nabla_x \mathcal{L}(g_\phi(f_\theta(\mathbf{x})), \mathbf{y})|| \tag{2}$$

The underlying premise is that the larger the loss gradient, the higher the dependence level. Previous studies also supports the rationality of gradient-based feature significance analysis. (Samek et al., 2017; Sun et al., 2021)

**Salient Input Feature Erasing.** After selecting the salient input features $\mathbf{x}_{\text{sf}}$ that the current mainline representation $\mathbf{z}_M$ most heavily depends on, the next step is to perform $\text{MASK}(\cdot)$ operation on the raw input $\mathbf{x}$. For image data, the masking operation works at the level of image patch and $\mathbf{x}_{\text{sf}}$ designates a certain proportion of image patches based on the Equation 2. We then fill these patches with random values that conform to a Gaussian distribution $\mathcal{N}(\mu, \sigma)$, where the $\mu$ and $\sigma$ are calculated based on empirical distribution of pixels of the whole training set. Additionally, we ensure that the padded pixel values are within the range of valid image through the clip operation.

As for text data, each raw input $\mathbf{x}$ is represented as a sequence of tokens, and thus the masking operation is performed at token level. As we utilize pretrained language models with special tokens, we erase text feature by replacing a certain proportion of tokens selected by gradient norm of token embeddings with [MASK] token.

Additionally, it should be emphasized that the modification on $\mathbf{x}$ ought to be kept within a restricted range, as we expect that the remaining part still contains relevant information about $\mathbf{y}$. The model could be negatively affected by excessive erasure since the obtained $\mathbf{x}'$ may lose excessive supervisory information.

## 2.3 THE THIRD STAGE: JOINT TRAINING OF MAINLINE AND SUPPLEMENTAL FEATURES

In the third stage, the task is to jointly train the whole model and simultaneously learn the mainline features $\mathbf{z}_M$ and the supplemental features $\mathbf{z}_S$. The mainline features $\mathbf{z}_M$ still uses the same supervised objective as the one used in the first stage (Equation 1), while the $\mathbf{z}_S$ needs to suppress learning salient features $\mathbf{x}_{\text{sf}}$ of $\mathbf{z}_M$ while learning supervised objective.

The intuitive idea here is to force the model to continuously acquire new potential features distinct from the features already learnt by current representation $\mathbf{z}_M$, and expect these new features to be helpful for classification. We refer to these features as supplementary features, and we think the key characteristic of these features is that they are partially non-overlapping features with respect to $\mathbf{x}_{\text{sf}}$

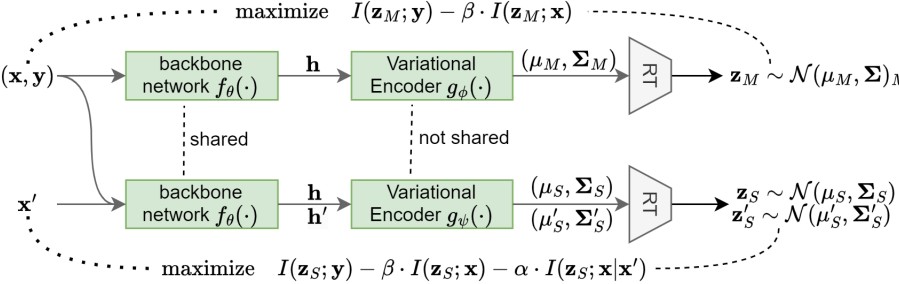

Figure 2: The architecture and information flow of the third stage for joint learning of mainline and supplemental features.

and are easy to be suppressed or overlooked by the mainline features $\mathbf{z}_M$, possibly weaker or harder, as long as they will contribute to the supervisory task.

To accomplish this, we leverage another supplementary representation $\mathbf{z}_S$ to learn these supplementary features. However, it is not guaranteed that $\mathbf{z}_S$ won't extract the same salient feature $\mathbf{x}_{\text{sf}}$ that $\mathbf{z}_M$ has already learnt if there are no constraints on $\mathbf{z}_S$. As a result, $\mathbf{z}_S$ should be restricted not to learn features that exists in $\mathbf{x}$ but not in $\mathbf{x}'$. According to the chain rule of mutual information, the information $\mathbf{z}_S$ contains about $\mathbf{x}$ could be divided into two parts(the proof can be found at Appendix B.1):

$$I(\mathbf{z}_S; \mathbf{x}) = I(\mathbf{z}_S; \mathbf{x}') + I(\mathbf{z}_S; \mathbf{x}|\mathbf{x}') \tag{3}$$

The first term represents the information $\mathbf{z}_S$ contains about $\mathbf{x}'$, and the second term intuitively represents the information $\mathbf{z}_S$ contains which is unique to $\mathbf{x}$ and is not predictable by observing $\mathbf{x}'$. This term is exactly what we tend to suppress as we only expect $\mathbf{z}_S$ to learn information relevant to $\mathbf{y}$ from $\mathbf{x}'$. In this way, we can derive the learning objective of $\mathbf{z}_S$ as follow:

$$\text{maximize} \quad I(\mathbf{z}_S; \mathbf{y}) - \beta \cdot I(\mathbf{z}_S; \mathbf{x}) - \alpha \cdot I(\mathbf{z}_S; \mathbf{x}|\mathbf{x}') \tag{4}$$

where the the first two terms represent supervised loss with information bottleneck, which is consistent with $\mathbf{z}_M$'s learning objective. The third item is a regularization restriction on $\mathbf{z}_S$ representation aiming to force it to learn supplementary features.

Figure 2 illustrates the architecture and information flow of the third stage. We use two different variational encoder $g_\phi$ and $g_\psi$ to generate $\mathbf{z}_M$ and $\mathbf{z}_S$, respectively. However, the backbone is shared, and the variational encoder only has very few parameters, so our method only uses a small number of additional parameters, which is different from model integration. Additionally, although the joint training stage makes use of the modified inputs generated by the second stage, their use in our method are essentially different from the other data-augmentation-based methods. It should be noted that the supervision loss for $\mathbf{z}_S$ is still calculated on the representation corresponding to the original inputs $\mathbf{x}$ but not the modified inputs $\mathbf{x}'$. The modified inputs $\mathbf{x}'$ are used only for a regularization term $I(\mathbf{z}_S; \mathbf{x}|\mathbf{x}')$. Therefore, in contrast to data-augmentation-based methods, we need not worry about the incorrect supervision signal (or wrong label) that are possibly introduced to the modified inputs $\mathbf{x}'$.

## 2.4 MODEL ARCHITECTURE AND LOSS FUNCTION

In this section, we specifically discuss how to achieve optimization goals proposed above in practice. From the perspective of architecture, we design a network with two different variational encoder head on a shared backbone. It should be noted that in inference stage, we will use both $\mathbf{z}_M$ and $\mathbf{z}_S$ by averaging logits.

As for the optimizing objectives of each stage, firstly, for the Equation 1, the loss could be derived as follow based on the variational estimate proposed by Alemi et al.:

$$\mathcal{L}_{\text{VIB}}(\mathbf{x}, \mathbf{z}_M, \theta, \phi) = \mathbb{E}_{\mathbf{x}}[\mathbb{E}_{\mathbf{z}_M \sim p_{\theta,\phi}(\mathbf{z}_M|\mathbf{x})}[-\log q(\mathbf{y}|\mathbf{z}_M)] + \beta \cdot D_{\text{KL}}[p_{\theta,\phi}(\mathbf{z}_M|\mathbf{x})||r_\phi(\mathbf{z}_M)]] \tag{5}$$

where $r_\phi(\mathbf{z}_M)$ is prior distribution of $\mathbf{z}_M$. We use a parameterized Gaussian distribution $\mathcal{N}(\mu_\phi, \Sigma_\phi)$ to represent $r_\phi(\mathbf{z}_M)$. As for $I(\mathbf{z}_S; \mathbf{x}|\mathbf{x}')$ used in Equation 4, we restrict the term by minimizing its upper bound (the proof can be found at Appendix B.2):

$$\mathcal{L}_{\mathrm{IS}} = \mathbb{E}_{\mathbf{x},\mathbf{x}'}[D_{\mathrm{KL}}[p_{\theta,\psi}(\mathbf{z}_S|\mathbf{x})||p_{\theta,\psi}(\mathbf{z}'_S|\mathbf{x}')]] \tag{6}$$

In this way, the total loss of the third stage can be derived as

$$\mathcal{L} = \mathcal{L}_{\mathrm{VIB}}(\mathbf{x}, \mathbf{z}_M, \theta, \phi) + \mathcal{L}_{\mathrm{VIB}}(\mathbf{x}, \mathbf{z}_S, \theta, \psi) + \alpha \cdot \mathcal{L}_{\mathrm{IS}} \tag{7}$$

where $\alpha$ is a coefficient that controls the weight of information suppression. Besides, since the distribution of both $\mathbf{z}_M$ and $\mathbf{z}_S$ follows a Gaussian distribution, there exists a closed form solution for calculating KL-divergence in the above losses.

## 3 EXPERIMENTS

This section is devoted to a thorough empirical study of the proposed InfoR-LSF method[1]. We first evaluate it in-domain generalization ability on classification and regression tasks (Section 3.1) and then investigate the out-of-domain generalization of learned representation on sentiment analysis task (Section 3.2). In addition, we also conduct experiments to analyze the effect of information retention (Section 3.3) and study the hyperparameter sensitivity (Section 3.4). Finally, results of ablation experiments are presented in Section 3.5.

**Architectures.** We choose ResNet-18 (He et al., 2016) as the backbone for image classification, the pretrained $\mathrm{BERT}_{\mathrm{BASE}}$ (Devlin et al., 2018) as the backbone for text-related tasks, and MLP network as the backbone for tabular regression. More dataset information and implementation details about training hyper-parameters and optimizer are provided in the Appendix A.1 and Appendix A.2.

**Baselines**. We compare against several influential works: **IFM** (Robinson et al., 2021), a method which avoids shortcut solutions by implicit feature modification; **FGSM** (Goodfellow et al., 2014), a classic adversarial training method in computer vision; **VIB** (Alemi et al., 2017), a variational approximation to the information bottleneck by leveraging the reparameterization trick; **VIBERT** (Mahabadi et al., 2021), a method implementing the variational information bottleneck on the pretrained BERT to suppress irrelevant features and enhance generalization ability when fine-tuning. In addition, the basic ResNet-18 (or $\mathrm{BERT}_{\mathrm{BASE}}$) model serves as a baseline.

### 3.1 IN-DOMAIN GENERALIZATION ON SUPERVISED LEARNING TASK

Table 1: CIFAR10 classification task accuracy under different train data size.

| Model | Train Data Size | | | | | | | |
|---|---|---|---|---|---|---|---|---|
| | 50 | 100 | 200 | 500 | 1000 | 2000 | 3000 | 50000 |
| ResNet-18 | 17.2 | 22.6 | 31.1 | 40.4 | 48.9 | 63.3 | 74.2 | 95.1 |
| IFM | 17.1 | 22.4 | 31.5 | 42.1 | 51.8 | 65.8 | 75.1 | 94.6 |
| FGSM | 20.1 | 23.7 | 31.4 | 40.3 | 47.7 | 58.1 | 65.5 | 91.8 |
| VIB | 18.6 | 22.4 | 31.0 | 39.7 | 49.9 | 64.8 | 74.7 | 95.1 |
| InfoR-LSF | **20.3** | **24.5** | **32.1** | **42.1** | **52.8** | **67.3** | **76.2** | **95.2** |
| $\Delta$ | +3.1 | +1.9 | +1.0 | +1.7 | +3.9 | +4.0 | +2.0 | +0.1 |

**Image classification.** We use two image classification datasets, CIFAR10 (Krizhevsky et al., 2009) and CIFAR100(Krizhevsky et al., 2009). Different sizes of training data (subsampled from the original training set), ranging from 50 to 50000, are used to evaluate the test accuracy in low-resource settings. Each experiment is conducted three times, each time with a different random seed for data sampling. The average test accuracy for CIFAR10 are shown in Table 1. The performance on CIFAR100 is provided in Appendix A.3.

**Text classification.** We use two sentiment analysis datasets, namely IMDB (Maas et al., 2011) and YELP (Zhang et al., 2015). The sizes of training data range from 50 to 1000, with training examples

---

[1]Code available at `https://github.com/liyahe/InfoR-LSF`

Table 2: Text classification task accuracy under different train data size.

| Dataset | Model | Train Data Size | | | | |
|---------|-------|------|------|------|------|------|
| | | 50 | 100 | 200 | 500 | 1000 |
| IMDB | BERT | 66.6 (2.2) | 77.9 (2.3) | 85.6 (0.5) | 87.1 (0.6) | 88.7 (0.3) |
| | IFM | 66.1 (2.2) | 78.2 (2.4) | 85.6 (0.7) | 87.4 (0.7) | 88.7 (0.4) |
| | VIBERT | 68.9 (2.5) | 80.8 (1.7) | 86.1 (0.6) | 87.8 (0.7) | 88.8 (0.4) |
| | InfoR-LSF | **75.5 (2.3)** | **83.0 (2.9)** | **86.9 (0.4)** | **88.3 (0.5)** | **89.4 (0.4)** |
| | $\Delta$ | +8.9 | +5.1 | +1.3 | +1.2 | +0.7 |
| YELP | BERT | 35.1 (1.8) | 39.6 (2.1) | 43.1 (1.7) | 51.9 (0.9) | 55.6 (0.7) |
| | IFM | 35.7 (2.5) | 40.1 (1.8) | 43.4 (1.0) | 50.9 (1.0) | 55.5 (0.7) |
| | VIBERT | 37.7 (1.2) | 40.8 (2.3) | 44.8 (2.2) | 53.1 (2.2) | 55.4 (0.6) |
| | InfoR-LSF | **39.6 (1.1)** | **41.4 (1.4)** | **44.9 (2.4)** | **53.6 (0.6)** | **55.9 (0.3)** |
| | $\Delta$ | +4.5 | +1.8 | +1.8 | +1.7 | +0.3 |

randomly sampled with five seeds (13, 21, 42, 87 and 100). We report the average and standard deviation of test accuracy in Table 2. Here, under low resource settings, we do not use the original validation set, but instead sample a validation subset of the same size as the train data.

Table 3: STS-B test set Pearson correlation coefficient under different train data sizes.

| Dataset | Model | Train Data Size | | | | |
|---------|-------|------|------|------|------|------|
| | | 50 | 100 | 200 | 500 | 1000 |
| STS-B | BERT | 72.2 (3.2) | 79.1 (1.9) | 83.8 (0.8) | 86.4 (1.0) | 87.5 (0.2) |
| | IFM | 72.3 (3.1) | 79.2 (1.9) | 84.0 (0.9) | 86.8 (0.7) | 87.6 (0.2) |
| | VIBERT | 74.4 (2.8) | 81.9 (1.8) | 85.0 (0.4) | 87.1 (0.3) | 88.4 (0.3) |
| | InfoR-LSF | **75.0 (3.1)** | **82.4 (2.0)** | **85.4 (0.5)** | **87.5 (0.6)** | **88.7 (0.3)** |
| | $\Delta$ | +2.8 | +3.3 | +1.6 | +1.1 | +1.2 |

**Textual Similarity Score Regression.** The task STS-B(Cer et al., 2017) is to regress textual similarity score. The results of STS-B are shown in Table 3.

**Traditional Tabular Regression.** We further conduct experiments on a tabular regression task Appliance Energy Prediction (AEP) (Candanedo, 2017) from UCI Machine Learning Repository (Asuncion & Newman, 2007). We process the data as same as public preprocessing [2] and then apply a 5-layer MLP for prediction. We adjust the hyper-parameters to make MLP reach the public SOTA performance, and then verify the effect of InfoR-LSF on this basis. The results of energy prediction are shown in Table 4.

Table 4: Coefficient of determination($R^2$) of AEP under different train data sizes.

| Model | Train Data Size | | | |
|-------|------|------|------|------|
| | 10% | 20% | 50% | 100% |
| MLP | 0.338 | 0.456 | 0.597 | 0.684 |
| IFM | 0.373 | 0.469 | 0.605 | 0.680 |
| VIB | 0.347 | 0.471 | 0.602 | 0.679 |
| InfoR-LSF | **0.376** | **0.483** | **0.618** | **0.691** |
| $\Delta$ | +0.038 | +0.027 | +0.021 | +0.007 |

**Observation and Analysis.** *Firstly*, from Table 1 and Table 2, it is obvious that our InfoR-LSF method surpasses all competitors under all settings of training data sizes, for both image and text classification tasks. Table 3 and Table 4 further prove that InfoR-LSF also performs well on regression tasks. These observations indicate the universality of information retention principle. We attribute this substantial performance gain to its ability of learning diverse features, including not only the mainline features but also the supplemental features (note that the supplemental features may be redundant with respect to the mainline ones from the perspective of training data). *Secondly*, VIB achieves little performance gain on CIFAR, while VIBERT exhibits more performance gain on text-related tasks. We attribute this to the essential difference between images and texts: the tokens, as the basic units of texts, are meaningful, and

---

[2]https://www.kaggle.com/code/rrakzz/r2-68-accuracy-95

only a few of them can indicate the label of a given text; however, the pixels, as the basic units of images, are meaningless alone, over-simple features that comprise fewer pixels may not necessarily imply better generalization ability. That also explains why VIB is more often applied to NLP tasks. *Last but not the least*, InfoR-LSF exhibits much notable improvements in low resource conditions, and its performance gain gradually declines as the number of training examples rises. It suggests that the feature redundancy may diminish with the increased availability of labeled data resource, and the train and test sets tend to be more equally-distributed as the data size grows.

## 3.2 OUT-OF-DOMAIN PERFORMANCE

Since InfoR-LSF is wedded to learn more versatile relevant features, it is naturally hypothesized that the learned representation is beneficial to similar out-of-domain tasks. To verify this hypothesis, we choose the sentiment classification task, and use the full-size YELP data (Zhang et al., 2015) as the source domain to train models. By freezing the backbone and retraining a linear task-specific classification head, we evaluate the linear readout of each model on a series of out-of-domain target tasks, including IMDB (Maas et al., 2011), YELP-2 (Zhang et al., 2015), SST-2 (Socher et al., 2013), SST-5 (Socher et al., 2013), MR (Pang & Lee, 2005), Amazon-2 (Zhang et al., 2015) and Amazon-5 (Zhang et al., 2015). Each linear head is trained with 1000 labeled data from the target task. As shown in Table 5, on all target tasks, InfoR-LSF consistently achieves the highest improvement. We conjecture the reason to be our method's ability of extracting more versatile features and thus the learned representation is more likely to cover the useful features in target domains, leading to better out-of-domain generalization.

Table 5: Test accuracy of models transferring to new target datasets. All models are trained on YELP and evaluated linear readout on the target datasets. $\Delta$ are the absolute differences with BERT.

| Model | Target Dataset | | | | | | | |
|---|---|---|---|---|---|---|---|---|
| | YELP | YELP-2 | IMDB | SST-2 | SST-5 | MR | Amazon-2 | Amazon-5 |
| BERT | 65.81 | 94.95 | 88.24 | 86.54 | 44.88 | 80.70 | 81.59 | 54.53 |
| VIBERT | 66.00 | 95.87 | 88.05 | 83.90 | 44.75 | 81.20 | 81.81 | 56.05 |
| InfoR-LSF | **66.31** | **95.89** | **88.55** | **88.19** | **46.28** | **82.00** | **83.03** | **57.43** |
| $\Delta$ | +0.5 | +0.94 | +0.31 | +1.65 | +1.4 | +1.3 | +1.44 | +2.9 |

## 3.3 EFFECT VERIFICATION OF INFORMATION RETENTION

The main motivation of our method is to extract as much information about $\mathbf{y}$ from $\mathbf{x}$ as possible. Previous experiments have demonstrated its superiority in various downstream tasks, but is the improvement in effect due to the idea of information retention? To explore this question, we investigate whether the model exploits more input features by observing the model's attention distribution on input images. If the learned representation at the final hidden layer utilizes more original input features, the attention of the model should be distributed more evenly on more pixels. Otherwise, the attention will be distributed intensively on fewer pixels. Therefore, we use the gradient norm of loss as the model's attention score for each pixel, which has been proven to be an effective method for evaluating the contribution of features (Simonyan et al., 2013; Zeiler & Fergus, 2014). Specifically, we first obtain the gradient $\nabla_x \mathcal{L}(\mathbf{z}, \mathbf{y})$ for each pixel $x \in \mathbf{x}$, normalize the absolute values of the gradients over the entire image, and then examine the distribution of gradient amplitudes across the entire test set. From the visualized results in Figure 3, it can be observed that our method has the most dispersed attention distribution on the original input $\mathbf{x}$ under different training data sizes, and its distribution peak is significantly lower than other methods. On the contrary, VIB has the highest peak of gradient distribution, and thus attends intensively on the fewest pixels. The phenomenon meets our expectations, as the VIB restricts the mutual information between the representation $\mathbf{z}$ and input $\mathbf{x}$ through an explicit regularization term, so the utilization of information by the representation $\mathbf{z}$ on $\mathbf{x}$ is insufficient. Surprisingly, The gradient distribution of FGSM, which is based on adversarial training, is actually more concentrated than baseline, indicating that adversarial attacks can improve the robustness of the model, but may not necessarily force the model to leverage more original input information. In summary, compared to other baselines, our method exhibits a more uniform and dispersed trend in pixel level attention distribution, demonstrating the mechanism of information retention is in work effectively.

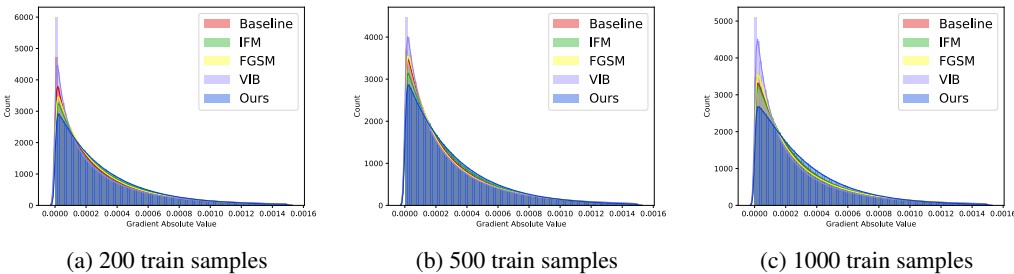

| (a) 200 train samples | (b) 500 train samples | (c) 1000 train samples |

Figure 3: Gradient amplitude distribution over test set of different method on CIFAR10.

## 3.4 SENSITIVITY ANALYSIS OF $\alpha$ AND MASKING RATIO

To examine the effect of loss term $\mathcal{L}_{\text{IS}}$ in Equation 7, We perform sensitivity analysis experiments by varying coefficient $\alpha$ and masking ratio on the IMDB dataset under low resource setting (50 train data). As shown in Figure 4a (or 4b), test and validation accuracy first increase and then decrease by increasing $\alpha$ (or by increasing masking ratio), and they both achieve the best performance at $\alpha = 1e-3$ (or with the masking ratio of 0.05). It means that over-high values of $\alpha$ and making ratio put excessive restriction on $\mathbf{z}_S$ representation, and thus do harm to the performance.

Figure 4c shows the difference in the attention distribution of $\mathbf{z}_S$ and $\mathbf{z}_M$ on salient input features $\mathbf{x}_{\text{sf}}$. We calculate the attention proportion by first calculating the gradient norm $||\nabla_{\mathbf{x}_{\text{sf}}}\mathcal{L}(\mathbf{z}_M, \mathbf{y})||$ and $||\nabla_{\mathbf{x}_{\text{sf}}}\mathcal{L}(\mathbf{z}_S, \mathbf{y})||$, and then normalizing them across the whole sentence. Obviously, the gap between $\mathbf{z}_M$ and $\mathbf{z}_S$ gradually increases, indicating that the restrictions on regularization terms are tightening. Additionally, we have noticed that an excessively large $\alpha$ will also constrain $\mathbf{z}_M$ meanwhile because $\mathbf{z}_M$ and $\mathbf{z}_S$ share the encoder parameter. This can also explain why excessive $\alpha$ is harmful.

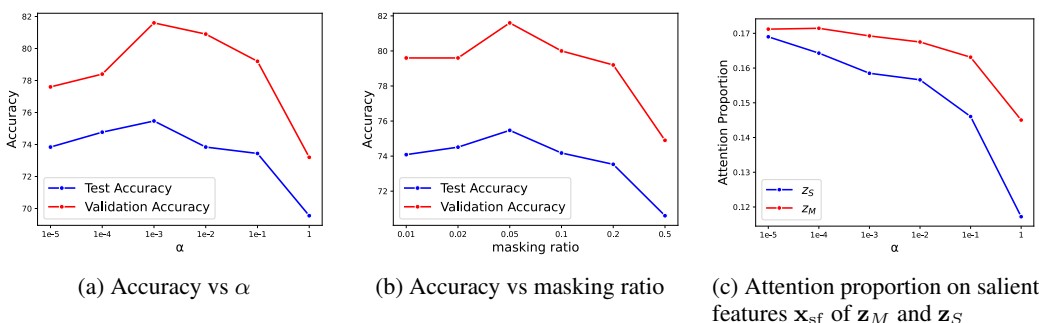

| (a) Accuracy vs $\alpha$ | (b) Accuracy vs masking ratio | (c) Attention proportion on salient features $\mathbf{x}_{\text{sf}}$ of $\mathbf{z}_M$ and $\mathbf{z}_S$ |

Figure 4: Sensitivity analysis of $\alpha$ and masking ratio on IMDB dataset.

## 3.5 ABLATION STUDY

To explore the effect of the regularization restriction on $\mathbf{z}_S$ and the IB restriction, we conduct ablation experiments on three datasets. We remove the regularization restriction on $\mathbf{z}_S$ by setting $\alpha = 0$ and remove IB restriction by setting $\beta = 0$. The results are shown in Table 6. *Firstly*, regardless of whether there are IB restrictions, the performance does reduce on all considered datasets if $\alpha = 0$, demonstrating the solid gain of the regularization term of our method. *Secondly*, we find that in some cases(especially on CIFAR10), only removing the IB restriction($\beta = 0$) results in performance improvement, indicating that the IB restriction can lead to negative effects in some cases. The phenomenon is also consistent with our previous analysis in Section 3.1 that the gains brought by IB on text are more significant than that on images. *Furthermore*, it should be noted that the ablation results still differ from the base model due to the use of two variational encoders. When $\alpha = 0$, InfoR-LSF degenerates to a VIB model with two heads (the modified input $\mathbf{x}'$ will not participate in any loss calculation if $\alpha = 0$). In this case, the results should be similar to VIB or VIBERT.

Table 6: Average ablation results over 5 runs under different data size of base model, InfoR-LSF and InfoR-LSF without regularization restrictions

| Data | Model | Train Data Size | | | | |
|------|-------|-----|-----|-----|-----|------|
| | | 50 | 100 | 200 | 500 | 1000 |
| IMDB | BERT | 66.6 (2.2) | 77.9 (2.3) | 85.6 (0.5) | 87.1 (0.6) | 88.7 (0.3) |
| | InfoR-LSF | **75.5 (2.3)** | **83.0 (2.9)** | **86.9 (0.4)** | **88.3 (0.5)** | **89.4 (0.4)** |
| | InfoR-LSF($\alpha$=0) | 73.6 (3.1) | 81.1 (3.9) | 86.0 (0.9) | 87.5 (0.4) | 88.8 (0.2) |
| | InfoR-LSF($\beta$=0) | 71.3 (2.8) | 80.9 (2.4) | 86.4 (0.5) | 87.8 (0.4) | 89.2 (0.2) |
| | InfoR-LSF($\alpha$=0,$\beta$=0) | 68.4 (2.7) | 78.5 (2.6) | 85.9 (0.9) | 87.5 (0.5) | 88.6 (0.6) |
| YELP | BERT | 35.1 (1.8) | 39.6 (2.1) | 43.1 (1.7) | 51.9 (0.9) | 55.6 (0.7) |
| | InfoR-LSF | **39.6 (1.1)** | **41.4 (1.4)** | 44.9 (2.4) | **53.6 (0.6)** | **55.9(0.3)** |
| | InfoR-LSF($\alpha$=0) | 37.8 (2.1) | 41.0 (1.4) | 43.6 (2.1) | 53.3(0.7) | 55.2 (0.9) |
| | InfoR-LSF($\beta$=0) | 38.6 (2.5) | 41.0 (3.6) | **45.6 (1.9)** | 53.5 (0.8) | 55.7 (0.5) |
| | InfoR-LSF($\alpha$=0,$\beta$=0) | 36.5 (1.7) | 40.1 (2.5) | 43.8 (2.3) | 51.3 (0.7) | 55.1 (0.3) |
| CIFAR10 | ResNet | 17.2(2.1) | 22.6(1.1) | 31.1(1.6) | 40.4(1.6) | 48.9(1.2) |
| | InfoR-LSF | 20.3(1.5) | **24.5(0.2)** | **32.1(1.8)** | 42.1(0.8) | **52.8(1.0)** |
| | InfoR-LSF($\alpha$=0) | 19.2(0.8) | 22.5(0.5) | 30.4(1.8) | 41.0(0.4) | 51.1(0.9) |
| | InfoR-LSF($\beta$=0) | **20.4(1.6)** | 23.8(0.9) | 31.8(2.0) | **42.7(0.8)** | 52.1(1.9) |
| | InfoR-LSF($\alpha$=0,$\beta$=0) | 18.7(2.2) | 21.5(0.4) | 30.6(1.6) | 41.5(0.7) | 50.4(1.5) |

## 4 RELATED WORK

**Information Bottleneck.** The information bottleneck(IB) (Tishby et al., 1999) was first proposed in traditional machine learning and then used to analyze and interpret the behavior of deep neural networks (Tishby & Zaslavsky, 2015; Shwartz-Ziv & Tishby, 2017). Later, Alemi et al. (2017) presented variational information bottleneck (VIB) to improve the learning of DNNs by optimizing a variational bound of IB objective. (Federici et al., 2020) extended the IB method to learn robust representations in unsupervised multi-view setting by minimizing superfluous information not shared by the two views. Additionally, IB were introduced to NLP tasks such as dependency parsing (Li & Eisner, 2019) and unsupervised sentence summarization (West et al., 2019). Mahabadi et al. (2021) applied VIB in the finetuning of pretrained BERT (Devlin et al., 2018) , showing good low-resource performance. Besides, IB also helped domain generalization(Ahuja et al., 2021; Du et al., 2020).

**Information Maximization.** The InfoMax principle (Linsker, 1988; Bell & Sejnowski, 1995) were proposed to advocate maximizing mutual information between input and output. Hjelm et al. (2019) applied InfoMax principle on unsupervised representations learning for deep neural networks and Bachman et al. (2019) further developed self-supervised representation learning based on maximizing mutual information between features extracted from multiple views. Besides, the InfoMax principle has also been leveraged in contrastive representation learning, such as contrastive predictive coding (Oord et al., 2018; Henaff, 2020) and contrastive multi-view coding (Tian et al., 2020).

Although it has been observed that IB methods can learn parsimonious features with better generalization in resource-limited scenarios, they implicitly suppress other redundant relevant features. The main motive of this work advocates the learned representations should incorporate these suppressed relevant features, which makes it distinct from IB methods.

## 5 CONCLUSION

In this work, we introduce the principle of information retention, which aims to keep as much relevant information as possible in use for making predictions. We further design a three-stage supervised learning framework named InfoR-LSF for information retention by jointly learning the mainline and supplemental features. In experiments, we compare InfoR-LSF against other methods, and its strong performance in different fields under both low-resource and out-of-domain scenario shows that InfoR-LSF can be practically applied to various type of tasks. Furthermore, analysis experiments indicate that our framework indeed achieves information retention and extracts more relevant features than other competitors.

ACKNOWLEDGMENTS

This work is financially supported by National Natural Science Foundation of China (No.62076072). We are grateful to the anonymous reviewers for their valuable comments.

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

# A  EXPERIMENTAL SUPPLEMENT

## A.1  DATASET STATISTICS

Table 7 shows statistic information of datasets used in our experiments.

Table 7: Datasets used in experiments.

| Dataset | #Lables | Train | Valid | Test |
|---|---|---|---|---|
| Image Classification | | | | |
| CIFAR10 | 10 | 50K | - | 10K |
| CIFAR100 | 10 | 50K | - | 10K |
| Sentiment Classification | | | | |
| IMDB | 2 | 20K | 5K | 25K |
| YELP | 5 | 62.5K | 7.8K | 8.7K |
| YELP-2 | 2 | 560K | - | 38K |
| SST-2 | 2 | 6.9K | 0.9K | 1.8K |
| SST-5 | 5 | 8.5K | 1.1K | 2.2K |
| MR | 2 | 8.7K | - | 2K |
| Amazon-2 | 2 | 3600K | - | 400k |
| Amazon-5 | 5 | 3000K | - | 650K |
| Semantic Textual Similarity | | | | |
| STS-B | 1 | 5.8K | 1.5K | 1.4K |
| Regression | | | | |
| Appliance Energy Prediction | 1 | 15.8K | - | 3.9K |

## A.2  IMPLEMENTATION DETAILS

For CIFAR10, refering to the open source implementation[3], we train the model for 200 epochs with batch size 256, initial learning rate 0.1 and weight decay 5e-4. We apply SGD optimizer with momentum 0.9 and a step scheduler that decays the learning rate by 0.1 every 160 epochs. For CIFAR100, we train the model for 240 epochs with batch size 64, initial learning rate 0.05 and weight decay 5e-4. We use SGD optimizer with momentum 0.9 and multi-step scheduler that decays the learning rate by 0.1 at milestones 150, 180, 210. For text classification tasks, we fine-tune all the models for 25 epochs with batch size 8 to allow them to converge. We use AdamW optimizer (Kingma & Ba, 2014) with initial learning rate 2e-5 and warmup ratio 0.05 for all models for fine-tuning. Besides, we use early stopping for in all text experiments based on the performance of the validation set.

As for hyper-parameters of variational information bottleneck, we use bottleneck size as half of the hidden layer representation dimension, i.e., 384 for BERT and 256 for ResNet-18. And the $\beta$ of information bottleneck is set to 1e-5 for all experiments. We use sample size 5 for reparameterization.

As for the coefficient $\alpha$ of our method, we use 1e-3 for all the text experiments, 1e-2 of CIFAR10 and CIFAR100 and 0.1 for Appliance Energy Prediction. And for the modification of input features, in all our experiments, the mask proportion is 5%, i.e., 5% of the patches in the image or 5% of the tokens for the sentence. In addition, for both CIFAR10 and CIFAR100, the patch size used when selecting salient features is $4 \times 4$, i.e., each image is divided into $8 \times 8$ image patches.

## A.3  IMAGE CLASSIFICATION PERFORMANCE ON CIFAR100

On CIFAR100, we construct low resource subset of various sizes from original train set, with scales ranging from 1000 to 50000. Results are provided in Table 8. Consistent with the conclusion on CIFAR10, our method outperforms all baseline methods at all data sizes.

---

[3]https://github.com/kuangliu/pytorch-cifar

Table 8: CIFAR100 classification task accuracy under different train data size.

| Model | Train Data Size | | | | | | |
|---|---|---|---|---|---|---|---|
| | **1000** | **2000** | **3000** | **5000** | **10000** | **20000** | **50000** |
| ResNet-18 | 13.90 | 20.65 | 27.10 | 38.08 | 55.52 | 67.14 | 77.85 |
| IFM | 14.04 | 21.71 | 28.46 | 39.46 | 56.72 | 67.19 | 77.53 |
| FGSM | 14.19 | 20.56 | 26.21 | 34.80 | 48.46 | 59.60 | 71.66 |
| VIB | 13.94 | 21.17 | 27.85 | 39.46 | 56.30 | 67.30 | 77.54 |
| InfoR-LSF | **15.51** | **22.61** | **30.43** | **43.32** | **58.79** | **68.85** | **78.44** |
| $\Delta$ | +1.61 | +1.96 | +3.33 | +5.24 | +3.27 | +1.71 | +0.59 |

### A.4 COMPUTATIONAL COMPLEXITY ANALYSIS

Theoretically, InfoR-LSF conducts forward propagation and backward propagation twice during the training phase, and there is double data in the second time. As a result, the time consumption should be approximately 2-3 times that of the base model. During the inference phase, InfoR-LSF doesn't require additional backward propagation.The increase in calculation amount only comes from the additional variational encoder. Therefore, the time cost should be comparable to the base model. We measure the running time of each method on CIFAR10 and IMDB. The results are shown in Table 9, in which the base model is used as a benchmark. It can be observed that the training time of InfoR-LSF is about 2.43 times that of the base model, and the inference time is about 1.03 times that of the base model, which is consistent with the conclusion of the theoretical analysis.

Table 9: Computational complexity analysis of each method. $\Delta\%$ are relative differences with base model(ResNet-18 or BERT).

| Dataset | Model | Train Time | $\Delta\%$ | Inference Time | $\Delta\%$ |
|---|---|---|---|---|---|
| CIFAR10 | baseline | 425s | - | 1.98s | - |
| | IFM | 430s | 1.18% | 2.02s | 1.97% |
| | FGSM | 859s | 102.12% | 1.99s | 0.66% |
| | VIB | 428s | 0.71% | 2.02s | 2.02% |
| | InfoR-LSF | 1033s | 143.06% | 2.05s | 3.28% |
| IMDB | baseline | 101s | - | 100.56s | - |
| | IFM | 105s | 4.16% | 101.54s | 0.98% |
| | VIBERT | 108s | 6.32% | 101.89s | 1.32% |
| | InfoR-LSF | 243s | 139.75% | 103.42s | 2.84% |

## B PROOFS

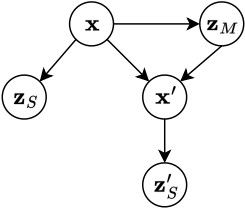

Figure 5: The relationship between $\mathbf{x}$, $\mathbf{x}'$, $\mathbf{z}_M$, $\mathbf{z}_S$, and $\mathbf{z}'_S$.

In the framework of InfoR-LSF, we first train $\mathbf{z}_M$ on original input $\mathbf{x}$, and then mask the salient feature of $\mathbf{x}$ according to $\mathbf{z}_M$ to obtain $\mathbf{x}'$. In the third stage, we jointly train both $\mathbf{z}_M$ and $\mathbf{z}_S$, where $\mathbf{z}_S$ is restricted by a regularization term. The relationship between $\mathbf{x}$, $\mathbf{x}'$, $\mathbf{z}_M$, $\mathbf{z}_S$, and $\mathbf{z}'_S$ is shown as Figure 5. It can be derived that $\mathbf{x}'$ and $\mathbf{z}_S$ are conditionally independent for any given $\mathbf{x}$.

## B.1 PROOF OF EQUATION 3

*Hypothesis:*

Given $\mathbf{x}$, $\mathbf{x}'$ and $\mathbf{z}_S$ are conditionally independent: $I(\mathbf{z}_S; \mathbf{x}'|\mathbf{x}) = 0$

*Proof.*

$$
\begin{aligned}
I(\mathbf{z}_S; \mathbf{x}) &= I(\mathbf{z}_S; \mathbf{x}\mathbf{x}') - I(\mathbf{z}_S; \mathbf{x}'|\mathbf{x}) \\
&= I(\mathbf{z}_S; \mathbf{x}\mathbf{x}') \\
&= I(\mathbf{z}_S; \mathbf{x}') + I(\mathbf{z}_S; \mathbf{x}|\mathbf{x}')
\end{aligned}
\tag{8}
$$

## B.2 PROOF OF EQUATION 6

*Hypothesis:*

Given $\mathbf{x}$, $\mathbf{x}'$ and $\mathbf{z}_S$ are conditionally independent: $p(\mathbf{z}_S\mathbf{x}'|\mathbf{x}) = p(\mathbf{z}_S|\mathbf{x})p(\mathbf{x}'|\mathbf{x})$

*Proof.*

$$
\begin{aligned}
I(\mathbf{z}_S; \mathbf{x}|\mathbf{x}') &= \mathbb{E}_{\mathbf{x}'\sim p(\mathbf{x}')}\mathbb{E}_{\mathbf{z}_S,\mathbf{x}\sim p(\mathbf{z}_S,\mathbf{x}|\mathbf{x}')} \log \frac{p(\mathbf{z}_S, \mathbf{x}|\mathbf{x}')}{p(\mathbf{z}_S|\mathbf{x}')p(\mathbf{x}|\mathbf{x}')} \\
&= \mathbb{E}_{\mathbf{x},\mathbf{x}'\sim p(\mathbf{x},\mathbf{x}')}\mathbb{E}_{\mathbf{z}_S\sim p(\mathbf{z}_S|\mathbf{x}\mathbf{x}')} \log \frac{p(\mathbf{z}_S, \mathbf{x}|\mathbf{x}')}{p(\mathbf{z}_S|\mathbf{x}')p(\mathbf{x}|\mathbf{x}')} \\
&= \mathbb{E}_{\mathbf{x},\mathbf{x}'\sim p(\mathbf{x},\mathbf{x}')}\mathbb{E}_{\mathbf{z}_S\sim p(\mathbf{z}_S|\mathbf{x}\mathbf{x}')} \log \frac{p(\mathbf{z}_S|\mathbf{x}\mathbf{x}')}{p(\mathbf{z}_S|\mathbf{x}')} \\
&= \mathbb{E}_{\mathbf{x},\mathbf{x}'\sim p(\mathbf{x},\mathbf{x}')}\mathbb{E}_{\mathbf{z}_S\sim p(\mathbf{z}_S|\mathbf{x})} \log \frac{p(\mathbf{z}_S|\mathbf{x})}{p(\mathbf{z}_S|\mathbf{x}')} \\
&= \mathbb{E}_{\mathbf{x},\mathbf{x}'\sim p(\mathbf{x},\mathbf{x}')}\mathbb{E}_{\mathbf{z}_S\sim p(\mathbf{z}_S|\mathbf{x})} \log \frac{p(\mathbf{z}_S|\mathbf{x})p(\mathbf{z}'_S|\mathbf{x}')}{p(\mathbf{z}'_S|\mathbf{x}')p(\mathbf{z}_S|\mathbf{x}')} \\
&= \mathbb{E}_{\mathbf{x},\mathbf{x}'\sim p(\mathbf{x},\mathbf{x}')}[D_{\text{KL}}[p(\mathbf{z}_S|\mathbf{x})||p(\mathbf{z}'_S|\mathbf{x}')] - D_{\text{KL}}[p(\mathbf{z}_S|\mathbf{x}')p(\mathbf{z}'_S|\mathbf{x}')]] \\
&\leq \mathbb{E}_{\mathbf{x},\mathbf{x}'\sim p(\mathbf{x},\mathbf{x}')}[D_{\text{KL}}[p(\mathbf{z}_S|\mathbf{x})||p(\mathbf{z}'_S|\mathbf{x}')]]
\end{aligned}
\tag{9}
$$

# C ALGORITHM TABLE OF INFOR-LSF

The framework of InfoR-LSF is shown in Algorithm 1.

---

**Algorithm 1** InfoR-LSF

---

**Input**: Input data $\mathbf{x} \in \mathcal{R}^M$ and label $\mathbf{y}$. A model $\mathcal{M}$ consisting with a backbone network $f_\theta(\cdot)$ and two variational encoders $g_\phi(\cdot)$ and $g_\psi(\cdot)$. Training epochs $T$. Feature suppression coefficient $\alpha$ and VIB coefficient $\beta$.
**Output**: Final model $\mathcal{M}$.
 1: Train mainline feature $\mathbf{z}_M = g_\phi(f_\theta(\mathbf{x}))$ with VIB loss $\mathcal{L}_{\text{VIB}}(\mathbf{x}, \mathbf{z}_M, \theta, \phi)$ at the first epoch.
 2: **for** $t = 1, 2, ...., T-1$ **do**
 3:     Find salient input feature of mainline feature $\mathbf{z}_M$ as $\mathbf{x}_{\text{sf}} = \text{topK}_{x\in\mathbf{x}} ||\nabla_x\mathcal{L}(g_\phi(f_\theta(\mathbf{x})), \mathbf{y})||$
 4:     Erase salient feature $\mathbf{x}_{\text{sf}}$ from $\mathbf{x}$ to obtain erased input $\mathbf{x}'$.
 5:     Jointly train mainline feature $\mathbf{z}_M = g_\phi(f_\theta(\mathbf{x}))$ and supplemental feature $\mathbf{z}_S = g_\psi(f_\theta(\mathbf{x}))$ by
        $\mathcal{L} = \mathcal{L}_{\text{VIB}}(\mathbf{x}, \mathbf{z}_M, \theta, \phi) + \mathcal{L}_{\text{VIB}}(\mathbf{x}, \mathbf{z}_S, \theta, \psi) + \alpha \cdot \mathcal{L}_{\text{IS}}$
 6: **end for**
 7: **return** Final model $\mathcal{M}$

---

# D MOTIVATIONS FOR INFORMATION RETENTION PRINCIPLE

## D.1 FROM THE PERSPECTIVE OF CAUSAL DIAGRAM

To motivate the information retention principle, let us investigate the causal diagram in Figure 6. Since the common cause **h** are latent and unobserved, correlation exists between the input **x** and the label $y$. Traditional machine learning or deep learning methods usually exploits the correlation to make prediction are based on the correlation, with $\hat{y} = f(\mathbf{x})$ as the predicted label for the input data **x**. At the level of causal mechanism, the task of inferring $y$ from **x** consists of two steps: to infer the posterior of latent cause **h** of the observed input **x**, and then to predict $y$ based on $p(\mathbf{h}|\mathbf{x})$. During this process, taking more input features of **x** into consideration is expected to yield better estimation of **h**, although some components of **h** may have no effect on $y$.

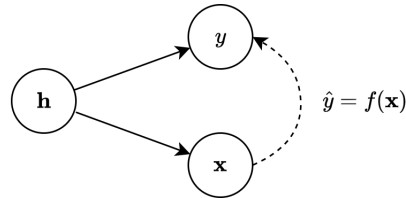

Figure 6: The causal diagram that assumes that there is a common cause **h** of the input **x** and its real label $y$

### D.2  A TOY EXAMPLE TO ILLUSTRATE THE IDEA OF INFORMATION RETENTION

Table 10: A simple motivating example

| $x_1$ | $x_2$ | $x_3$ | $x_4$ | $y$ |
|---|---|---|---|---|
| 1 | 1 | 1 | 2 | True |
| 1 | 1 | 1 | 2 | True |
| 0 | 2 | 2 | 2 | True |
| 0 | 3 | 2 | 2 | False |
| 1 | 2 | 2 | 1 | False |
| 0 | 3 | 2 | 1 | False |

Table 10 shows a simple classification dataset, where the input consists of four variables $\{x_1, x_2, x_3, x_4\}$ and the label $y$ is a binary variable. We use this example to illustrate the idea of information retention principle.

The label $y$ can be perfectly predicted by using the feature $f_1 = x_1 + x_2$, because $f_1 = 2$ implies $y = True$, and $f_1 = 3$ implies $y = False$. Therefore, the feature $f_1$ can be thought of as a good representation, which is both concise and predictive.

Next, let us have a look at the other two simplest features $f_2 = x_3$ and $f_3 = x_4$, which are also relevant to the label and have medium predictive ability. However, since $f_1$ has captured all the information of the label $y$, taking $f_2$ or $f_3$ into consideration will not bring any lifting in predictive ability but will lead to increased mutual information with the input. Thus, both $f_2$ and $f_3$ are suppressed and discarded by the information bottleneck principle.

Finally, it comes to the test time and a new data $[x_1 = 1, x_2 = 3, x_3 = 1, x_4 = 2]$ arrives, which has the feature $f_1 = 4$. The classification model based on $f_1$ has not seen this feature value in the training phase and cannot make a reliable prediction of label. However, feature $f_2$ and feature $f_3$ can deal with this situation, and both of them supports the prediction of $y = True$.

Therefore, it is beneficial and desirable to endow with the ability of relieve redundant relevant features from the suppression of existing mainline features. Such redundant relevant features is supplemental to the mainline features.

Furthermore, given the model that heavily relies on $f_1$ (called the mainline feature) and thus attends to the input features $x_1$ and $x_2$, one possible solution to build up some new supplemental features by requiring them to capture the least information that has already been captured by mainline features, and simultaneously maximize their mutual information with the label.

## E  AN EXAMPLE FOR VISUALIZATION OF LEARNED FEATURES

Figure 7 shows an example from IMDB test set. We compute the gradient norms on input token embedding sequences and normalize them over the whole sentence. The three lines respectively represent the gradient distribution of BERT, $\mathbf{z}_M$ head and $\mathbf{z}_S$ head of InfoR-LSF. The depth of the color represents the magnitude of the gradient. It can be observed that with the regularization term $\mathcal{L}_{\text{IS}}$, $\mathbf{z}_S$ partly erases salient features of $\mathbf{z}_M$("movie a 1 .") and learns new features("made me sick .") which are also helpful for classification. In addition, from the perspective of out-of-domain generalization, "movie a 1 ." is a task-specific feature while "made me sick" is a more general feature, this can also explain the better out-of-domain performance of InfoR-LSF.

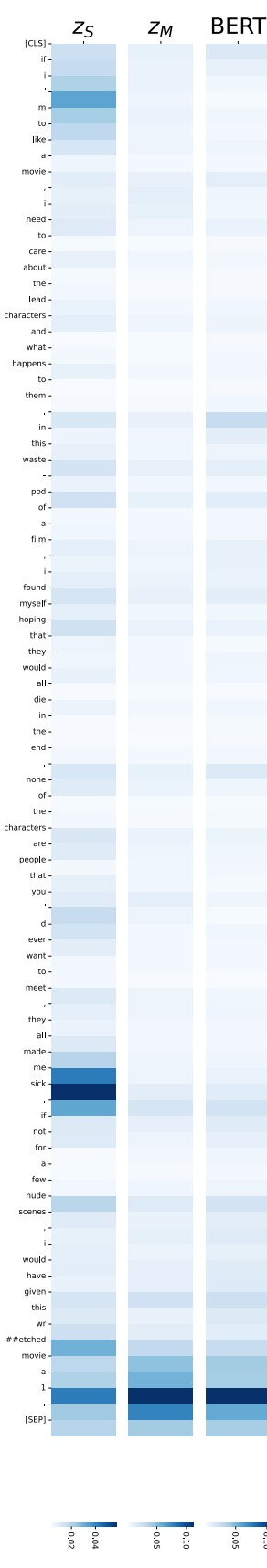

Figure 7: Visualization of one example on IMDB test set. The depth of the color represents the magnitude of the gradient.

