# OpenReview forum: "Information Retention via Learning Supplemental Features"
_ICLR.cc/2024/Conference — ICLR 2024 spotlight_

### Official Review · Reviewer_GgyT · 2023-10-30

**Soundness:** 4 excellent
**Presentation:** 3 good
**Contribution:** 3 good
**Rating:** 8
**Confidence:** 4

**Summary:**

This paper proposes a new paradigm of feature learning: Information Retention. The idea is to extract and keep as much *relevant* information as possible. This differs from the Information Bottleneck as such that Information Bottlenect tries to find the most compact informative features -- and this one does not focus on compactness, but keep as many relevant features. The proposed method consists of three stages that reflect the idea: (1) extract main features (2) find the most salient features -- make a copy of features with the salient ones removed (3) joint training of the features found in (1) and (2).

**Strengths:**

- The paper is well written
- The writers did a good job to soundly motivate why it makes sense to retain as much relevant feature as possible -- going slightly against the common paradigm to fins the most compact feature representation
- Reasonable baselines and works on multiple modality
- I like that the authors perform verification experiment (3.3); this strengthen the claim.

**Weaknesses:**

- One of my biggest concern is: in the abstract, the motivation was to enable better ood/distribution shift prediction. but none of the evaluation reflects this. in fact, the motivation seem to focus on low-resource settings. Does the authors focus is to generalize from learning on a small sample? If yes, this should be made clearer in the writing.
- An algorithm table would help a lot in method clarity

**Questions:**

1. In Weaknesses
2. Why does the feature erasure done in sample space, and not in feature space?

---

> ### Author Response · Authors · 2023-11-20
>
> Thanks for your careful and valuable comments. We will explain your concerns point by point.
>
> **Weakness  1**: We believe that our proposed InfoR-LSF has the ability to generalize outside the domain since InfoR-LSF is wedded to learn more versatile relevant features. To verify this hypothesis, we choose the sentiment classification task, and use the full-size YELP data as the source domain to train models. By freezing the backbone and retraining a linear task-specific classification head, we evaluate the linear readout of each model on a series of out-of-domain target tasks, including IMDB, YELP-2, SST-2, SST-5, MR, Amazon-2 and Amazon-5. Each linear head is trained with 1000 labeled data from the target task. As shown in the table below, on the all target tasks, InfoR-LSF consistently achieves the highest improvement. We conjecture the reason to be our method's ability of extracting more versatile features and thus the learned representation is more likely to cover the useful features in target domains, leading to better out-of-domain generalization. In addition, you can also refer to Section 3.2 of the submitted revised paper.
>
> | Model             | Target Dataset |           |           |           |           |           |           |           |
> | ----------------- | :------------: | :-------: | :-------: | :-------: | :-------: | :-------: | :-------: | :-------: |
> |                   |      YELP      |  YELP-2   |   IMDB    |   SST-2   |   SST-5   |    MR     | Amazon-2  | Amazon-5  |
> | BERT              |     65.81      |   94.95   |   88.24   |   86.54   |   44.88   |   80.70   |   81.59   |   54.53   |
> | VIBERT            |     66.00      |   95.87   |   88.05   |   83.90   |   44.75   |   81.20   |   81.81   |   56.05   |
> | InfoR-LSF         |   **66.31**    | **95.89** | **88.55** | **88.19** | **46.28** | **82.00** | **83.03** | **57.43** |
> | $\mathbf{\Delta}$ |      +0.5      |   +0.94   |   +0.31   |   +1.65   |   +1.4    |   +1.3    |   +1.44   |   +2.9    |
>
>
>
> **Weakness  2**: We provide an algorithm table in Appendix C, which we hope will help you better understand the framework of InfoR-LSF.
>
>
>
> **Question 1**: Please refer to answer of Weaknesses.
>
> **Question 2**:
> Yes, in our work, feature erasing is done at the level of input features, instead of the level of representation features.
>
> If we try to do feature erasing in feature space, one straightforward way is to mask some mainline features. However, this operation do not necessarily relieve supplemental features out, but instead possibly do learn duplicate mainline features that are masked.
>
> Another way is to minimize the mutual information between the mainline features and the supplemental features, as questioned by the first reviewer. We copy the answer below:
>
>  We do not directly minimize the mutual information between $\mathbf{z}_M$ and $\mathbf{z}_S$ for two reason. On the one hand, our motive is to relieve the supplemental features $\mathbf{z}_S$ (which are possibly redundant relevant features) from the suppression of mainline features $\mathbf{z}_M$. $\mathbf{z}_S$ and $\mathbf{z}_M$ may be mutually redundant on the training set, that is, they can be different features with high mutual information between each other. Minimizing the mutual information between $\mathbf{z}_M$ and $\mathbf{z}_S$ will also suppress the supplemental features $\mathbf{z}_S$ and cannot achieve our goal. On the other hand, in our framework,  both  $I(\mathbf{z}_M, \mathbf{y})$  and $I(\mathbf{z}_S, \mathbf{y})$  should be maximized, and this conflicts with minimizing the $I(\mathbf{z}_M, \mathbf{z}_S)$. While $I(\mathbf{z}_S;\mathbf{x}|\mathbf{x}')$ represents the information $\mathbf{z}_S$ contains which is unique to $\mathbf{x}$ and is not predictable by observing $\mathbf{x}'$, the goal is to let $\mathbf{z}_S$ learn features different from $\mathbf{z}_M$ .

---

> > ### Comment · Reviewer_GgyT · 2023-11-20
> >
> > Thank you for the rebuttal! My concern is addresses, and I have changed my score to accept

---

### Official Review · Reviewer_WVdm · 2023-10-31

**Soundness:** 3 good
**Presentation:** 2 fair
**Contribution:** 3 good
**Rating:** 8
**Confidence:** 2

**Summary:**

The paper presents a novel concept in supervised learning called information retention, emphasizing the importance of utilizing as much relevant information as possible for predictions. The authors introduce the InfoR-LSF framework, a three-stage learning system designed to process both mainline and supplemental features without allowing the mainline features to suppress the supplemental ones. Through experiments on the CIFAR10, IMDB, and YELP datasets, the paper demonstrates that InfoR-LSF outperforms existing methods, especially in low-resource scenarios. This research underscores the potential of InfoR-LSF to be effectively applied across various tasks, highlighting its unique approach to retaining and utilizing information.

**Strengths:**

1. The introduction of the concept of information retention and the development of the InfoR-LSF framework offer a fresh perspective on supervised learning. The method's focus on harnessing both mainline and supplemental features without interference distinguishes it from conventional approaches.

2. The paper conducts experiments on multiple datasets spanning different domains (CIFAR10 for image classification and IMDB and YELP for text classification). This extensive evaluation offers credibility to the method's versatility and robustness across varying types of data and tasks.

3. Beyond just presenting a novel method and its performance metrics, the paper delves deep into understanding its functioning. Through experimental verifications, like observing the model's attention distribution on input images, the authors effectively demonstrate the actual retention of information, strengthening the paper's central claim.

**Weaknesses:**

1. While the paper evaluates performance on both image and text classification tasks, it doesn't expand into other types of tasks (e.g., regression, segmentation, or sequence-to-sequence tasks), potentially limiting the generalizability of the findings.

2. The three-stage supervised learning framework has several components. An ablation study detailing the contribution of each component to the final performance would have given insights into the necessity and utility of each part.

3. The paper doesn’t discuss any potential increase in computational complexity or training time introduced by the InfoR-LSF framework, which can be critical for real-world applications.

4. Apart from the sensitivity analysis of coefficient α, the paper does not deeply dive into how sensitive the model is to other hyperparameters, which can be crucial for replication and understanding model robustness.

**Questions:**

1. Why were CIFAR10, IMDB, and YELP selected as the benchmark datasets? Are there plans to test the method on more diverse or domain-specific datasets?

2. How generalizable is the InfoR-LSF framework? Can it be easily adapted to other tasks or domains outside of image and text classification?

3. Given that adversarial training methods like FGSM were considered as baselines, did the authors consider evaluating the adversarial robustness of models trained with InfoR-LSF?

---

> ### Author Response · Authors · 2023-11-20
>
> Thanks for your careful and valuable comments. We will explain your concerns point by point.
>
> **Weakness 1**:
> We believe that our proposed InfoR-LSF can be expanded to other types of tasks. Due to time constraints, we only conduct experiments on two regression tasks, including the semantic similarity dataset STS-B and a tabular dataset Appliance Enengy Prediction from UCI ML Repository. The experimental results are shown in the tables below(you can also find the results in Section 3.1 of the submitted revised paper):
>
> STS-B
> | Models            | Train Data Size |               |               |                |                |
> | :---------------- | :-------------: | :-----------: | :-----------: | :------------: | :------------: |
> |                   |       50        |      100      |      200      |      500       |      1000      |
> | BERT              |    72.2(3.2)    |   79.1(1.9)   |   83.8(0.8)   |   86.4(1.0)    |   87.5 (0.2)   |
> | IFM               |    72.3(3.1)    |   79.2(1.9)   |   84.0(0.9)   |   86.8(0.7)    |   87.6 (0.2)   |
> | VIBERT            |    74.4(2.8)    |   81.9(1.8)   |   85.0(0.4)   |   87.1 (0.3)   |   88.4 (0.3)   |
> | InfoR-LSF         |  **75.0(3.1)**  | **82.4(2.0)** | **85.4(0.5)** | **87.5 (0.6)** | **88.7 (0.3)** |
> | $\mathbf{\Delta}$ |      +2.8       |     +3.3      |     +1.6      |      +1.1      |      +1.2      |
>
> ApplianceEnergyPrediction
> |      Models       | Train Data Size |           |           |           |
> | :---------------: | :-------------: | :-------: | :-------: | :-------: |
> |                   |       10%       |    20%    |    50%    |   100%    |
> |        MLP        |      0.338      |   0.456   |   0.597   |   0.684   |
> |        IFM        |      0.373      |   0.469   |   0.605   |   0.680   |
> |        VIB        |      0.347      |   0.471   |   0.602   |   0.679   |
> |     InfoR-LSF     |    **0.376**    | **0.483** | **0.618** | **0.691** |
> | $\mathbf{\Delta}$ |     +0.038      |  +0.027   |  +0.021   |  +0.007   |
>
> The above results prove that InfoR-LSF also performs well on regression tasks.
>
> Furthermore, we believe that our method should be effective on seq2seq tasks. For example, in NER, the entity word and its context are two different features. The vanilla method has been shown to work by simply memorizing the entity words, which leads to poor out-of-vocabulary generalization. Our method should be able to adaptively learn the contexts as supplemental features and elegantly solve the problem.
>
> As to image segmentation, we are uncertain whether our approach would work or not.
>
> **Weakness 2**:
> We update the original ablation experiment and conduct more specific ablation experiments on the hyperparameters $\alpha$ and $\beta$. Please refer to Section 3.5 of the submitted revised paper for experimental results and conclusions.
>
> **Weakness 3**:
> We have analyzed the computational cost from both theoretical and experimental perspectives. The results are shown in Appendix A.4 of the submitted revised paper. Roughly, the training time of InfoR-LSF is about 2.43 times that of the base model, and the inference time is about 1.03 times. This is acceptable in real-world applications.
>
> **Weakness 4**:
> We update the analysis experiments and add analysis on masking ratio, another important hyperparameter of our method. For specific experiments, please refer to Section 3.4 of the submitted revised paper.
>
> **Question 1**:
> We choose CIFAR10 because it was commonly used for image classification. We have also made experiment on CIFAR100, and the results are put in Appendix A.3 of the submitted revised paper. As to text classification, we choose IMDB and YELP, just because their texts are relatively long and likely to contain redundant relevant features.
> In addition, we have applied our method on two regression datasets, please refer to answer of Weakness 1 for more details.
>
> **Question 2**:
> The principle of information retention is general, and the framework for learning supplemental features is also easy to adapt to other tasks. But some technical particularities need attention, such as the saliency erasing mechanisms.
> Actually, we have adapted InfoR-LSF to regression tasks as in the answer of Weakness 1.
>
> **Question 3**:
> The reason of using FGSM as baseline is that FGSM also erases the learned features from the original examples and thus generates adversarial examples, which is somewhat similar to our approach.
> The difference is that FGSM uses the adversarial examples as data augmentation which require the augmented examples be of the same labels as the original examples; but our approach uses the modified example to exert the constraint of conditional mutual information $I(\mathbf{z}_S;\mathbf{x}|\mathbf{x}')$ as in Equation (4), which does not suffer from the label-consistency limitation of data augmentation.
> Currently, we have not made any attempts to check the adversarial robustness of InfoR-LSF, but we think it could be a future direction.

---

> > ### Comment · Reviewer_WVdm · 2023-11-22
> > **Thanks for the rebuttal**
> >
> > I thank the authors for addressing my comments. My concerns are largely addressed, so I have increased my rating.

---

### Official Review · Reviewer_LTAN · 2023-10-31

**Soundness:** 3 good
**Presentation:** 2 fair
**Contribution:** 3 good
**Rating:** 6
**Confidence:** 3

**Summary:**

The paper presents a new model architecture based on the intuition that redundant yet relevant features are also crucial, contradicting the information bottleneck principle. In detail, the authors designed a three-stage training algorithm: firstly, to learn the mainline feature; secondly, to erase the salient input; and thirdly, to learn the supplementary feature. The authors conducted classification tasks in both vision and language, and performed analyses to validate the usage of redundant features.

**Strengths:**

The paper starts with the intuition that the information bottleneck principle may be validated under specific situations and then formulates the idea into a training pipeline and model architecture. This is a very exciting buildup from idea to implementation. At the same time, the authors design the whole pipeline in a way that is very close to the initial idea, and they consider the details very thoughtfully. In the analysis part, they also provide sufficient evidence that the redundant yet relevant features are truly learned.

**Weaknesses:**

1. The presentation can be greatly improved. For instance, out-of-domain scenarios are mentioned in the abstract, but they are not explored in later experiments. The two examples in the introduction are unnecessary; they might even hinder comprehension, as some context is missing (as in example 1.1), and the usage could be restricted by the examples provided.
2. Section 3.3 shows the proposed algorithm extracts more information than other algorithms. It is also necessary to show that the extracted information is actually relevant information.
3. It would be helpful to provide some visualization to show that the second stage does erase the salient features.
4. Some context in information theory is missing. A section of related work is needed.

**Questions:**

How is Equation (5) and (6) exactly derived?

---

> ### Author Response · Authors · 2023-11-20
>
> Thanks for your careful and valuable comments. We will explain your concerns point by point.
>
> **Weakness 1**:Thank you for your suggestion. We have checked the writing and moved the two examples from Section 1 to the Appendix D in the submitted revised version. In addition, we have provided out-of-domain experiments in the revised version. Implementation and results are as follow:
>
> We believe that the learned representations of InfoR-LSF have the ability to generalize outside the domain since InfoR-LSF is wedded to learn more versatile relevant features. To verify this hypothesis, we choose the sentiment classification task, and use the full-size YELP data as the source domain to train models. By freezing the backbone and retraining a linear task-specific classification head, we evaluate the linear readout of each model on a series of out-of-domain target tasks, including IMDB, YELP-2, SST-2, SST-5, MR, Amazon-2 and Amazon-5. Each linear head is trained with 1000 labeled data from the target task. As shown in the table below, on the all target tasks, InfoR-LSF consistently achieves the highest improvement. We conjecture the reason to be our method's ability of extracting more versatile features and thus the learned representation is more likely to cover the useful features in target domains, leading to better out-of-domain generalization. In addition, you can also refer to Section 3.2 of the submitted revised paper.
>
> | Model             | Target Dataset |           |           |           |           |           |           |           |
> | ----------------- | :------------: | :-------: | :-------: | :-------: | :-------: | :-------: | :-------: | :-------: |
> |                   |      YELP      |  YELP-2   |   IMDB    |   SST-2   |   SST-5   |    MR     | Amazon-2  | Amazon-5  |
> | BERT              |     65.81      |   94.95   |   88.24   |   86.54   |   44.88   |   80.70   |   81.59   |   54.53   |
> | VIBERT            |     66.00      |   95.87   |   88.05   |   83.90   |   44.75   |   81.20   |   81.81   |   56.05   |
> | InfoR-LSF         |   **66.31**    | **95.89** | **88.55** | **88.19** | **46.28** | **82.00** | **83.03** | **57.43** |
> | $\mathbf{\Delta}$ |      +0.5      |   +0.94   |   +0.31   |   +1.65   |   +1.4    |   +1.3    |   +1.44   |   +2.9    |
>
> **Weakness 2**:
> We have tested the accuracy of the classification head attached to $\mathbf{z}_S$, and it is even slightly higher than the head of $\mathbf{z}_M$, which manifests the learned supplemental features are relevant to the target label. PS. The cross entropy loss can be thought of as a lower bound of mutual information.
>
> **Weakness 3**: We provide a visual example in Appendix E of the submitted revised paper. In the visual example, we present the gradient distribution of BERT, $\mathbf{z}_M$ head and $\mathbf{z}_S$ head of InfoR-LSF on a test case of IMDB. The depth of the color represents the magnitude of the gradient and it can be observed that $\mathbf{z}_S$ turly erases salient features of $\mathbf{z}_M$ ("movie a 1 .") and learns new features ("made me sick .") which are also helpful for classification.
>
> **Weakness 4**: We have added related work in the submitted revised paper.
>
> **Question 1**: Equation (5) is the loss function of the variational information bottleneck. Please refer to the paper "Alexander A. Alemi, Ian Fischer, Joshua V. Dillon, Kevin Murphy (2017) Deep variational information bottleneck" for derivation. The exact derivation of Equation (6) is provided in Appendix B of the submitted revised paper.

---

> > ### Comment · Reviewer_LTAN · 2023-11-23
> > **Response to the reviewer**
> >
> > Thank you for the response! It has resolved most of my concerns. I still have some questions here.
> >
> > Response to weakness 2: Where can I find the comparison of $z_s$ and $z_m$? Can you elaborate more?
> >
> > Response to weakness 3: This visualization is really interesting! Besides an example with strong explicit signal "give movie a 1" for sentiment classification, I also wonder what would the gradient distribution be like for a sentence with mostly implicit signal, which is more common in the IMDB dataset (as what I observed here https://huggingface.co/datasets/imdb).
> >
> > Generally speaking, I really like how the authors formulate the problem, and how they bring insights from information theory. I found the paper to be more notable for its thought-provoking ideas than its practical applications, which is reflected in the strengths I wrote. In terms of experimental results, the current application is limited to simple 2-way, 5-way, or 10-way classification tasks. While it doubles the training time for VIB and VIBERT, it only offers marginal improvements over them in these simple tasks. As such, I could not see scalability potential to more complex tasks in the future. Therefore, from my perspective, the practical applicability of the proposed technique is neither a major factor contributing to its merit at this stage, nor the criteria to change my recognition of the paper. However, I would like to see more robust evidence supporting the ideas' main and supplementary features as I mentioned above.

---

> > > ### Author Response · Authors · 2023-11-23
> > >
> > > Thanks for your comment about the work. We shall try to elaborate more about the comparison between $\mathbf{z}_S$ and $\mathbf{z}_M$ and the gradient distributions for sentences with mostly implicit signals.
> > >
> > > Since the rebuttal phase is to be ended about one and a half hour later, it seems we have no chance to accomplish the elaboration and post it here, but we shall try to make the correponding revision  in the final version once accepted for publication.

---

### Official Review · Reviewer_BzTN · 2023-11-01

**Soundness:** 1 poor
**Presentation:** 1 poor
**Contribution:** 1 poor
**Rating:** 6
**Confidence:** 4

**Summary:**

This paper proposes retaining supplemental features during training, as mainline features learned under the information bottleneck principle are insufficient to address out-of-distribution cases.

**Strengths:**

The illustration of the proposed method is easy to follow.

**Weaknesses:**

- The paper is unqualified to be a top conference paper in several aspects, including novelty, contribution, and writing quality.
- The experiment design is outdated as of 2023.
- The writing needs improvement. For example, in the abstract, the authors mention, "relevant features may be supplemental to the mainline features of the model." However, the subsequent phrase "to address this problem" lacks coherence. Additionally, in the conclusion, verb tenses are inconsistent, alternating between "introduce" and "proposed."
- The paper lacks a comprehensive literature review.

**Questions:**

- What contributions can the research in this paper make to the current mainstream deep learning models, such as GPT and Diffusion?

---

> ### Author Response · Authors · 2023-11-20
>
> Thanks for your careful and valuable comments. We will explain your concerns point by point.
>
> **Weakness1:**
> We would like to summarize the contribution and novelty here:
> 1) At the principle level, we propose Information Retention, to use as much relevant information of the input as possible in making prediction.
>
> 2) At the technical level, we designed a three-stage framework which is able to extract diverse features by relieving supplemental features from the suppression of mainline features.
>
> 3) At the experimental level, we conducted a variety of experiments to verify the effectiveness of the proposed InfoR-LSF method. Its learned representations have exhibited good in-domain performance (especially in the resource-limit scenarios) and good out-of-domain generalization ability.
>
> **Weakness2:**
> We have added a number of experiments in the submitted revised paper. It includes:
> 1) Experiments to validate the out-of-domain generalization of the learned representation
> 2) Experiments on two regression tasks (one for text regression, and the other for tabular regression)
> 3) Experimental results to report the computational costs.
> 4) Sensitivity analysis of the main hyperparameters
> 5) Ablation analysis to indicate the effectiveness of proposed components.
>
> **Weakness3:**
> We are sorry for the typos and error of spelling and syntax in the submitted paper. Thanks for pointing it out. We have checked the paper thoroughly, and make the corresponding modification to the submitted revised paper.
>
> **Weakness4:**
> We have added related work to the revised paper.
>
> **Question 1:**
> We think our approach could be applied if there are supervised signals or labels used. We also believe it has wide scope of applicability. In building large language models, the pretext tasks are usually self-supervised and it is possible to get our approach involved in. However, due to  resource limitation, we do not have a chance or plan to do such work. But we plan to study how to adapt our approach to self-supervised contrastive representation learning on images.

---

> ### Comment · Reviewer_BzTN · 2023-11-21
>
> Thank you for the comprehensive explanation. The revisions made to this paper have addressed its most critical issues. Consequently, I am revising my evaluation and raising the score to 6

---

### Official Review · Reviewer_RFaU · 2023-11-06

**Soundness:** 4 excellent
**Presentation:** 3 good
**Contribution:** 3 good
**Rating:** 8
**Confidence:** 4

**Summary:**

This paper introduces the principle of information retention, which stands somewhat on the opposite side of the principles of Occam’s razor and information bottleneck. It focuses on utilizing as much relevant information as possible in decision making instead of favoring simplest model. Next, this underlying idea get materialized by relieving supplemental features from the suppression of mainline features in a three-stage algorithmic framework. Extensive experimental results have manifested the effectiveness of the proposed method in both image classification and text classification tasks.

**Strengths:**

1) The approach overall is well-motivated and well-described. I like the organizational form of the paper, from conceptual idea to framework design and to concrete implementation, which is easy and natural to follow.
2) The principle of information retention and the method of learning supplemental features are novel and inspiring. It seems that the approach and method can be readily extended to other tasks.
3) Experimental results on image classification and text classification tasks have manifested the approach has good in-domain generalization ability, especially in poor-resource settings.

**Weaknesses:**

1) The method can be better described with an overall workflow or architecture.
2) The paper needs further refinement of some minor details, and I have found some grammar and spelling errors in the sections of experiments and conclusions.
3) Although I believe that the proposed approach can benefit out-of-domain generalization,it would have been better if some preliminary experiments had been conducted.

**Questions:**

1) Besides in-domain generalization, I wonder whether the proposed method has better out-of-domain generalization ability in some degree?
2) To my understanding, the supplemental features are learned by minimizing the conditional mutual information $I(\mathbf{x}, \mathbf{z}_S|\mathbf{x}’)$, is it right? Why not just minimizing the mutual information between $\mathbf{z}_S$ and $\mathbf{z}_M$?

**Details Of Ethics Concerns:**

No ethics  problem.

---

> ### Author Response · Authors · 2023-11-20
>
> Thanks for your careful and valuable comments. We will explain your concerns point by point.
>
> **Weakness 1**:
> In order to better understand the workflow of our method, we provide an algorithm table in Appendix C and a variable relationship diagram in Appendix B in the submitted revised paper. We hope this can help you clearly understand our method.
>
> **Weakness 2**:
> Thank you for your suggestion. We have carefully checked for grammar and spelling errors and corrected them in the submitted revised paper.
>
> **Weakness 3**:
> We believe that the learned representations of InfoR-LSF have the ability to generalize outside the domain since InfoR-LSF is wedded to learn more versatile relevant features. To verify this hypothesis, we choose the sentiment classification task, and use the full-size YELP data as the source domain to train models. By freezing the backbone and retraining a linear task-specific classification head, we evaluate the linear readout of each model on a series of out-of-domain target tasks, including IMDB, YELP-2, SST-2, SST-5, MR, Amazon-2 and Amazon-5. Each linear head is trained with 1000 labeled data from the target task. As shown in the table below, on the all target tasks, InfoR-LSF consistently achieves the highest improvement. We conjecture the reason to be our method's ability of extracting more versatile features and thus the learned representation is more likely to cover the useful features in target domains, leading to better out-of-domain generalization. In addition, you can also refer to Section 3.2 of the submitted revised paper.
>
> | Model             | Target Dataset |           |           |           |           |           |           |           |
> | ----------------- | :------------: | :-------: | :-------: | :-------: | :-------: | :-------: | :-------: | :-------: |
> |                   |      YELP      |  YELP-2   |   IMDB    |   SST-2   |   SST-5   |    MR     | Amazon-2  | Amazon-5  |
> | BERT              |     65.81      |   94.95   |   88.24   |   86.54   |   44.88   |   80.70   |   81.59   |   54.53   |
> | VIBERT            |     66.00      |   95.87   |   88.05   |   83.90   |   44.75   |   81.20   |   81.81   |   56.05   |
> | InfoR-LSF         |   **66.31**    | **95.89** | **88.55** | **88.19** | **46.28** | **82.00** | **83.03** | **57.43** |
> | $\mathbf{\Delta}$ |      +0.5      |   +0.94   |   +0.31   |   +1.65   |   +1.4    |   +1.3    |   +1.44   |   +2.9    |
>
> **Question 1**:
>  Please refer to answer of Weakness 3.
>
> **Question 2**:
> This is a good question. We do not directly minimize the mutual information between $\mathbf{z}_M$ and $\mathbf{z}_S$ for two reason. On the one hand, our motive is to relieve the supplemental features $\mathbf{z}_S$ (which are possibly redundant relevant features) from the suppression of mainline features $\mathbf{z}_M$. $\mathbf{z}_S$ and $\mathbf{z}_M$ may be mutually redundant on the training set, that is, they can be different features with high mutual information between each other. Minimizing the mutual information between $\mathbf{z}_M$ and $\mathbf{z}_S$ will also suppress the supplemental features $\mathbf{z}_S$ and cannot achieve our goal. On the other hand, in our framework,  both  $I(\mathbf{z}_M, \mathbf{y})$  and $I(\mathbf{z}_S, \mathbf{y})$  should be maximized, and this conflicts with minimizing the $I(\mathbf{z}_M, \mathbf{z}_S)$. While $I(\mathbf{z}_S;\mathbf{x}|\mathbf{x}')$ represents the information $\mathbf{z}_S$ contains which is unique to $\mathbf{x}$ and is not predictable by observing $\mathbf{x}'$, the goal is to let $\mathbf{z}_S$ learn features different from $\mathbf{z}_M$ .

---

> > ### Comment · Reviewer_RFaU · 2023-11-21
> >
> > Thanks for the rebuttal that makes a clear reply to my questions. Accordingly, I will raise the score.

---

### Author Response · Authors · 2023-11-23

Dear Reviewers (RFaU, BzTN, LTAN, WVdm, and GgyT),

We would like to express our sincere gratitude for your selfless hard work, giving us many valuable comments, helping us to better improve our papers, and also promoting our further thinking on some issues, and inspiring us to work in the future.

Best Regards,

Authors of Paper 4712

---

### Meta-Review · Area_Chair_yvzs · 2023-12-09

**Metareview:**

Based on the intuition that redundant yet relevant features are also crucial, this paper presents novel ideas for information retention and learning supplemental features. In particular, a three-stage learning framework has been developed to jointly learn mainline and supplemental features. Experimental results on the tasks from multiple domains under different scenarios validated the effectiveness of the proposed method, and also demonstrated its wide applicability. After the rebuttal, all the reviewers were very positive and agreed on the novelty and the generalizability of the proposed method, as well as the comprehensive evaluation.

**Justification For Why Not Higher Score:**

Currently, all the reviewers are very positive with scores 6/6/8/8/8 (average is 7.2). They all agree on the novelty and the generalizability of the proposed method, as well as the comprehensive evaluation. Therefore, I would like to recommend Accept with spotlight.

The potential scalability issue and other minor issues raised from reviewer LTAN were not addressed in the current version. That is why I did not recommend Accept with oral.

**Justification For Why Not Lower Score:**

The average score (7.2) of this paper is high enough. That is why I recommend Accept with spotlight, instead of Accept with poster.

---

### Decision · Program_Chairs · 2024-01-16

Accept (spotlight)